# Membrane Permeability Is Required for the Vasodilatory Effect of Carbonic Anhydrase Inhibitors in Porcine Retinal Arteries

**DOI:** 10.3390/ijms24098140

**Published:** 2023-05-02

**Authors:** Thor Eysteinsson, Andrea García-Llorca, Arnar Oessur Hardarson, Daniela Vullo, Fabrizio Carta, Claudiu T. Supuran

**Affiliations:** 1Department of Physiology, BioMedical Center, Faculty of Medicine, University of Iceland, IS101 Reykjavik, Iceland; 2Sezione di Scienze Farmaceutiche e Nutraceutiche, Neurofarba Department, University of Florence, Via Ugo Schiff 6, 50019 Florence, Italy

**Keywords:** carbonic anhydrase, myography, arterial wall tension, membrane impermeable, porcine

## Abstract

It has been demonstrated previously that a variety of carbonic anhydrase inhibitors (CAIs) can induce vasodilation in pre-contracted retinal arteriolar segments although with different efficacy and potency. Since the CAIs tested so far are able to permeate cell membranes and inhibit both intracellular and extracellular isoforms of the enzyme, it is not clear whether extra- or intracellular isoforms or mechanisms are mediating their vasodilatory effects. By means of small wire myography, we have tested the effects of four new CAIs on wall tension in pre-contracted retinal arteriolar segments that demonstrably do not enter cell membranes but have high affinity to both cytosolic and membrane-bound isoforms of CA. At concentrations between 10^−6^ M to 10^−3^ M, none of the four membrane impermeant CAIs had any significant effect on arteriolar wall tension, while the membrane permeant CAI benzolamide (10^−3^ M) fully dilated all arteriolar segments tested. This suggests that CAI act as vasodilators through cellular mechanisms located in the cytoplasm of vascular cells.

## 1. Introduction

Carbonic anhydrase inhibitors (CAIs) are used to treat elevated intraocular pressure in glaucoma [1,2] but can also affect retinal oxygenation, vascular tone, and blood flow [3,4]. We have found previously that CAIs with different structures [5] and lipophilicity [6] can act as potent vasodilators on isolated, pre-contracted segments of retinal arteries. The exact mechanism through which carbonic anhydrase exerts its effects on arterial wall tension and, thus, vascular tone, retinal oxygenation, and blood flow is still unknown. The basic physiological processes involved are of importance since the normal function of the retinal vasculature ensures a sufficient supply of oxygen and nutrients to photoreceptors and retinal neurons and, thus, vision. In addition, retinal vascular function is affected in a variety of diseases involving the retina, such as diabetic retinopathy, retinopathy of prematurity, retinal vascular occlusions, age-related macular degeneration, and glaucoma [7,8]. In some cases, treatment of diseases involving the retinal vasculature has been directed towards the processes regulating vascular tone in these vessels, either directly or indirectly [8,9]. Understanding the role of CA in regulating vascular tone and wall tension in the retinal vasculature and the mechanisms involved may, thus, lead to new treatments for these diseases [10]. Several hypotheses on the role of the enzyme in vascular function, and through what mechanism it exerts its effects on the vascular contractile state in retinal arterioles in health and disease have been proposed and tested. Most of these have so far been shown by the evidence to be insufficient or wrong. Changes in extracellular pH by CA inhibition have been ruled out as a contributing factor since CAIs can still induce vasodilation when pH is maintained at 7.4 outside the vessel walls [4,11]. It has been shown that intracellular acidification of smooth muscle cells in the walls of porcine retinal arterioles is increased by CAIs, but no relation between vascular relaxation and intracellular acidosis in these vessels was found [12]. There is evidence that the vascular endothelium plays a role to some extent since inhibition of the formation of nitric oxide (NO) by L-NAME, a NO synthase inhibitor, or blockage of the NO receptor guanylyl cyclase by 1H-[1,2,4]oxadiazole[4,3-a]quinoxalin-1-one (ODQ) reduces the vasorelaxation induced by CAIs; however, it does not fully block that effect [12]. It has been suggested that perhaps mechanisms other than CA inhibition are responsible for the vasorelaxant effects of CAIs on arterioles [11,13,14], although it is not clear by which mechanisms. These suggestions were put forward based on experiments showing that acidosis, hypercapnia, and the nominal removal of the substrates CO_2_ and HCO_3_^−^ of the carbonic anhydrase reaction had no effect on the vasodilation induced by CAIs on porcine retinal arteriole segments [11], and that relatively high concentrations of CAIs approved for clinical use (sulfonamides, such as acetazolamide or dorzolamide) were required to evoke vasodilation in pre-contracted arterioles [11]. However, we have found that some sulfonamide-based CAIs have a very low EC_50_ for vasodilation of retinal arterioles and varying potency depending on factors, such as their structure [5,6]. Inhibition of the enzyme is, therefore, likely to be a critical step, but it may be that a mechanism other than the action of CA as an enzyme is responsible for its regulatory role of arterial wall tension. Given that there are several isoforms of the CA and that these isoforms are located either inside the cytoplasm of cells or on the cell membrane, it is possible that the mechanisms involved in the regulation of vascular tone may be localized in the cytosol, the cell membrane, or both. Separating these possibilities would provide further information about the mechanisms involved. To that end, we have examined the effects of CAIs on arteriolar wall tension that cannot pass through cell membranes, are positively charged, and have a bis-guanidine moiety that induces highly polar features in them and, thus, do not act on the CA isoforms in the cytosol, in comparison with a CAI, that we have previously established is a potent vasodilator [5]. We find that CAIs must be membrane permeable to dilate pig retinal arteries, and that membrane-impermeable CAIs do not cause vasodilation. This suggests that the CA isoforms involved in the regulation of vascular tone are located in the cytoplasm of vascular cells.

## 2. Results

### 2.1. Myography

In this study, five compounds that function as carbonic anhydrase inhibitors were tested by small vessel myography on pre-contracted porcine retinal arteriolar segments. Four of these compounds were found to be membrane impermeable using standard methods to assess the penetrability of compounds through cell membranes [15]. A fifth compound, benzolamide, has long been considered a membrane impermeant sulfonamide CAI and is still frequently used as such, but it has been found to penetrate the cell membranes of red blood cells quite efficiently to a similar extent as acetazolamide [15]. We have shown previously that benzolamide is a potent vasodilator of pre-contracted retinal arteriolar vessel segments [5], and, thus, was used for comparison with the membrane-impermeable compounds, and to ensure that CAIs can in fact dilate the arteriolar vessel segment tested. We first applied the prostaglandin analog U-46619 (10^−6^ M) to the vessel to induce vasoconstriction, as shown in Figure 1, in all the experiments. Once the wall tension in the vessel wall had reached a plateau the first dose of the CAI to be tested was added. As shown in Figure 1A, 2,4,6-trimethyl-1-(4-sulfamoylphenethyl)pyridin-1-ium perchlorate, compound CAI **1**, an established membrane impermeant CAI [16,17], was added to the tissue bath in cumulative doses from 10^−6^ M to 10^−3^ M, but it did not have any significant effect on the wall tension of the vessel.

Similar results were found from all vessels tested with CAI **1** (n = 8; Figure 1D). A single dose of 10^−3^ M benzolamide was then added to the tissue bath which induced a significant (*p* > 0.05), immediate, and complete vasodilation, showing that a membrane permeant and broad CAI can dilate the vessel while the membrane-impermeant CAI does not. In all experiments, a single dose of 10^−3^ M benzolamide was used as control in the same manner with invariably the same result, i.e., complete vasodilation. The membrane impermeant CAI **2** was added to a different vessel in cumulative doses ranging between 10^−6^ M to 10^−3^ M, as shown in Figure 1B, which had no significant effects on the wall tension measured from the vessel. Essentially the same results were found in all vessels tested with CAI **2** (n = 8; Figure 1D). Figure 1C shows a continuous recording from a retinal arteriolar segment that was pre-contracted with 10^−6^ M U-46619 added at the time point indicated by a vertical arrow. Once the wall tension had reached a plateau the membrane impermeant CAI **3** was added to the bath in cumulative doses ranging between 10^−6^ M to 10^−3^ M, but the compound had no significant effect on the wall tension of the vessel wall at any concentration tested, while 10^−3^ M benzolamide induced a complete and rapid vasodilation in the vessel. Comparable results were obtained from all vessel segments tested with **3** (n = 8; Figure 1D). Figure 1D shows the mean dilation (±SEM) as a function of the concentration of all the compounds tested, including the membrane impermeant CAI **4** (n = 8) which was tested in the same manner as the other membrane impermeant CAIs. As shown in Figure 1D, none of the membrane impermeant CAIs had any significant effect on wall tension, while benzolamide induced vasodilation in a dose-dependent manner as shown previously [5].

### 2.2. Synthesis of Inhibitors of the Carbonic Anhydrases

CAIs **2** and **4** were obtained similarly to CAI **3** [18] by addition of the commercially available sufanilamide and metanilamide to cyanoguanidine in *n*-butanol using a stoichiometric amount of hydrochloric acid solution in water (Figure 1).

All synthesized compounds were purified by crystallization from isopropyl alcohol (IPA) and were fully characterized by means of ^1^H-NMR, ^13^C-NMR, and mass spectra (ESI-MS). The newly synthesized compounds were structured to be positively charged to ensure that they are membrane impermeable. The presence of a bis-guanidine moiety induces highly polar features in these compounds as well as the possibility of being doubly protonated at a physiological pH (at the two guanidine moieties).

### 2.3. In Vitro Carbonic Anhydrase Inhibition

The final compounds **2**–**4** were investigated in vitro for their ability to inhibit the physiologically most relevant CA isoforms I, II, IX, and XII by means of the stopped-flow technique [19] and compared to the standard CAI of the sulfonamide type acetazolamide (**AAZ**). The activities are all reported as K_I_ values in Table 1.

The reported compound CAI **3** showed high affinity for the tumor-related CA isoforms IX and XII with K_I_ values of 20.2 and 1.7 nM, respectively. Although CAI **3** resulted in a CA IX inhibitor as potent as the reference **AAZ** (K_I_s of 20.2 and 25.7 nM for **3** and **AAZ,** respectively) it resulted in 3.4-fold more effective when the XII isoform was considered (Table 1). As for the ubiquitously expressed hCA I and II isozymes, the associated K_I_ values of 4435 and 501 nM were obtained, and these make this compound not a valuable inhibitor for such isoforms. The shorter analog of CAI **3**, (i.e., compound **2**) resulted in a far more effective inhibitor on the CA panel considered in this study. For instance, data in Table 1 shows that the hCAs I and II were inhibited at concentration values of 361.0 and 134.6 nM, respectively. Noteworthy is that low nanomolar values were reported for the IX and XII isoforms (K_I_s of 1.43 and 0.5 nM, respectively). A similar kinetic trend was obtained for the *meta*-substituted regioisomer CAI **4**. Specifically compounds CAI **2** and **4** were almost equally potent inhibitors of either the tumor-associated isoform IX and XII, whereas a consistent value discrepancy of two-fold was observed when the associated cytosolic expressed K_I_s were compared with the *para*-regioisomer being the most effective (Table 1).

## 3. Discussion

It has now been well established that a broad range of CAIs can act as vasodilators on pre-contracted retinal arterioles [4,6,11,12]. This includes CAIs that are used clinically as pressure-lowering drugs to treat glaucoma [4] as well as newer compounds with different molecular structures and affinity for the different CA isoforms [5] and lipophilicity [6]. Although neither selective affinity for specific isoforms nor lipophilicity fully determines the potency of CAIs as vasodilators, the evidence suggests that cytosolic isoforms of carbonic anhydrase or cytosolic binding sites of membrane-bound isoforms play a critical role in mediating the effects of carbonic anhydrase on vascular wall tension. Herein we report a group of carbonic anhydrase inhibitors (i.e., CAIs **1**–**4**) that cannot enter cell membranes and have no significant effects on the wall tension of pre-contracted pig retinal arterial segments, while the CAI benzolamide that has been shown to enter red blood cells [15] and that has high affinity for the membrane-bound CA IV isoform is a potent vasodilator of these same vessels. It was long held that benzolamide was a membrane-impermeable compound and exerted its actions on the membrane-bound CA IV only, but the evidence indicates that its actions are cytosolic to a great extent. These findings demonstrate that it is important to assess the lipid solubility and the ability to penetrate cell membranes of new compounds acting as CA ligands to understand their mode of physiological action. Furthermore, they suggest that the mechanism by which CAIs act as vasodilators is located in the cytoplasm of vascular cells. It is known that CAIs can induce inhibition of the enzyme through at least four different inhibition mechanisms and binding modes [20,21]: (1) binding to the Zinc (Zn(II)) ion on the enzyme active site, the classic inhibition mechanism; (2) anchoring of the inhibitor to the zinc-coordinated water molecule within the enzyme active site; (3) occluding the entrance of the active side of the enzyme; (4) binding outside the active side cavity of the enzyme in a hydrophobic pocket adjacent to the entrance of the active site [22]. We do not know yet which of these inhibition mechanisms are of importance to the vasodilatory effects of CAIs or, indeed, if all or only some of them play a role. However, clearly, the binding to the enzyme or the active site that leads to changes in vascular wall tension must be localized within the cytoplasm, although the isoforms involved may be in the cell membrane or in the cytoplasm. CAIs that are selective for any of the four inhibition mechanisms or binding modes listed above could shed further light on the processes involved.

## 4. Materials and Methods

### 4.1. Myography Experiments

The preparation of the retinal arteriole segments and the small vessel myography technique used in this study have been described previously in detail [5,6]. In short, porcine eyes were obtained from a local abattoir and quickly transported to the laboratory in an oxygenated physiological saline solution (PSS) kept at 4 °C. The same PSS was used during transporting the eyes, during isolation of the retinal arteriolar segments, and as the bath solution during myography recordings. The PSS had the following composition (in mM): 112.6 NaCl; 5.91 KCl; 24.9 NaHCO_3_; 1.19 MgCl_2_; 1.18 NaH_2_PO_4_; 2.0 CaCl_2_; 11.5 glucose, all dissolved in a double distilled water. All PSS solutions were oxygenated with an air mixture of 95% O_2_ and 5% CO_2_, and continuously in the myography tissue baths. The pH of the PSS was maintained at 7.4. Eyes were bisected at the equator with a razor blade, and the anterior segment and the lens were removed. The vitreous was then gently removed by pushing it out of the eyecup with forceps. The posterior segment was then filled slowly with an oxygenated PSS. The retina and its vasculature were then examined under a dissecting stereoscope and an arteriolar segment of no more than 2 mm in length with a reasonably straight alignment and near the optic disk chosen. 

The segment was dissected with an incision on each end, and retinal tissue of about 1 mm in width on each side of the vessel was left intact. A short Tungsten wire (25 µm in diameter) was then inserted into the lumen of the vessel segment from one end to the other. The vessel segment with the wire inside the lumen was then placed in a DMT630MA wire myograph system (DMT A/S, Aarhus, Denmark) recording chamber and mounted in the system to record wall tension continuously. The vessel segment and tungsten wire were placed between two metal jaws in the recording chamber, the wire was clamped with the jaws, and each end of the wire was fastened with a screw to the jaw attached to a micrometer in the recording chamber. A second short tungsten wire with the same diameter was then carefully inserted into the lumen of the vessel segment along the top of the first wire and attached with screws to the force transducer jaw in the chamber. The chamber was then put under the stereoscope and a micrograph of the vessel segment was taken through it for exact measurement of its length. The DMT630 myograph system had four recording chambers with a liquid volume of 10 mL maximum in each chamber and a force transducer with jaws in each. Four separate vessel segments could, thus, be experimented on separately at the same time. An automatic buffer filler system (625FS, DMT A/S, Aarhus, Denmark) was attached to the DMT630 system to remove the PSS from the tissue baths by suction and refill them again quickly. The myograph system has a heating unit and a thermal probe, and this was used to maintain the temperature of the PSS in the chambers constant at 37 °C.

Continuous recordings of wall tension in all four chambers, and the temperature of the PSS as monitored by the thermal probe were acquired with LabChart Pro as a five-channel continuous record (ADInstruments, Oxford, UK). Once the temperature in the recording baths had reached 37 °C, the vessel segments were allowed to stabilize in there for 30 min. The wall tension normalization procedure was then carried out in accordance with the manufacturer’s (DMT A/S) protocol, taking into account the length of each segment as described in detail previously [5,6]. Experiments on the vessel segments were then initiated, and first, the vessels were pre-contracted with the thromboxane A2 analog U-46619 (9,11-dideoxy-9 a,11a methanoepoxy prostaglandin F_2_a) (Cayman Chemicals Inc, Tallinn, Estonia) and dissolved in double distilled water to yield a stock solution of 10^−3^ M. When adding U-46619 to the recording bath from the stock solution of 10^−3^ M, the final concentration in the 10 mL bath was 10^−6^ M. The maximum increase in wall tension induced by 10^−6^ M U-46619 varied between segments, but if contractions were below 1.0 mN/mm in an experiment, then the results were discarded. Once the increased wall tension had reached a peak which usually took several minutes, the CAI compound that was to be tested in each case was added to the bath. All of the CAIs tested in this study were dissolved in DMSO to stock solutions of 0.1 M and then diluted to the final concentration desired in the bath of the recording chamber. For each compound examined, concentration–response curves were generated from all vessel segments tested by adding first the compound to the bath to a final concentration of 10^−6^ M as the lowest value in all experiments and then adding the next dose to increase the final concentration at that point (e.g., 2 × 10^−6^ M) up to a cumulative final concentration of 10^−3^ M as the highest dose tested. At least a minute passed between increases in concentration.

### 4.2. Data Analysis

The traces of continuous myography recordings were transferred as data files from the original LabChart Pro recording files to SigmaPlot 13 (Systat Software Inc., San Jose, CA, USA) for analysis and graphic presentation, and all graphs presented were generated in SigmaPlot 13. All statistical analysis was conducted using GraphPad PRISM 9 (GraphPad Software Inc., San Diego, CA, USA). All results are presented as means ± SEM in this study.

### 4.3. Chemistry

Solvents and all reagents were purchased from Sigma-Aldrich, VWR, and TCI. Nuclear magnetic resonance (^1^H NMR, ^13^C NMR) spectra were recorded using a Bruker Advance III 400 MHz spectrometer in DMSO-d_6_. Chemical shifts are reported in parts per million (ppm) and the coupling constants (J) are expressed in Hertz (Hz). Splitting patterns are designated as follows s, singlet; d, doublet; t, triplet; m, multiplet; brs, broad singlet; dd, double of doubles. The assignment of exchangeable protons (NH) was confirmed by the addition of D_2_O. Analytical thin-layer chromatography (TLC) was carried out on Merck silica gel F-254 plates. Flash chromatography purifications were performed on Merck silica gel 60 (230–400 mesh ASTM) as the stationary phase, and ethyl acetate, *n*-hexane, acetonitrile, and methanol were used as eluents. The solvents used in MS measurements were acetone, acetonitrile (Chromasolv grade) purchased from Sigma-Aldrich (Milan, Italy) and mQ water 18 MΩ, obtained from Millipore’s Simplicity system (Milan, Italy). The mass spectra were obtained using a Varian 1200 L triple quadrupole system (Palo Alto, CA, USA) equipped with electrospray source (ESI) operating in both positive and negative ions. Stock solutions of analytes were prepared in acetone at 1.0 mg mL^−1^ and stored at 4 °C. Working solutions of each analyte were freshly prepared by diluting stock solutions in a mixture of mQ H_2_O/ACN 1/1 (*v*/*v*) up to a concentration of 1.0 μg mL^−1^ The mass spectra of each analyte were acquired by introducing, via syringe pump at 10/L min^−1^, the working solution. Raw data were collected and processed by Varian Workstation, version 6.8, software. All compounds reported here are >96% of purity by NMR.

General synthetic procedure for compounds **2**–**4**:

The syntheses of compounds **2**–**4** were performed according to the previously reported experimental procedure [18]. Specifically, the appropriate arylsulfonamide derivative **2a**–**4a** (0.5 g, 1.0 equiv.) was treated with cyanoguanidine **5** (1.0 equiv.) in *n*-butanol (5 mL) followed by the addition of 6 M aqueous HCl (1.0 equiv.). The reaction mixture was heated to 100 °C under stirring for 6 h, then the solvents were removed under vacuum to afford a residue which was crystallized from isopropyl alcohol (IPA) to afford the desired products as white solids.

4-(3-Carbamimidoylguanidino)benzenesulfonamide hydrochloride salt **2**

40% yield. ^1^H-NMR (400 MHz, DMSO) δ 10.13 (s, 1H, exchangeable with D_2_O), 7.72 (d, J = 8.9 Hz, 2H), 7.59–7.41 (m, 6H, 4H exchangeable with D_2_O), 7.26 (s, 2H, exchangeable with D_2_O), 7.17 (s, 2H, exchangeable with D_2_O). ^13^C-NMR (100 MHz, DMSO) δ 162.5, 155.3, 143.1, 138.8, 127.4, 120.5. Experimental are in agreement with the reported literature [23].

4-(2-(3-Carbamimidoylguanidino)ethyl)benzenesulfonamide hydrochloride salt **3**

Yield 50%. ^1^H-NMR (400 MHz, DMSO) δ 8.07 (brs, 3H, exchangeable with D_2_O), 7.78 (d, J = 8.3 Hz, 2H), 7.46 (d, J = 8.3 Hz, 2H), 7.34 (s, 2H, exchangeable with D_2_O), 3.12–3.02 (m, 2H), 2.97 (q, J = 6.7 Hz, 2H). ^13^C-NMR (100 MHz, DMSO) δ 163.9, 143.6, 142.5, 130.2, 126.9, 119.4, 40.4, 33.6. Experimental are in agreement with the reported literature [18].

3-(3-Carbamimidoylguanidino)benzenesulfonamide hydrochloride salt **4**

58% yield. ^1^H-NMR (400 MHz, DMSO) δ 9.98 (s, 1H, exchangeable with D_2_O), 7.83 (d, J = 1.3 Hz, 1H), 7.59 (m, 1H), 7.55–7.32 (m, 7H, 5H exchangeable with D_2_O), 7.13 (s, 2H, exchangeable with D_2_O). ^13^C NMR (100 MHz, DMSO) δ 162.6, 155.7, 145.5, 140.3, 130.2, 124.6, 121.1, 118.3.

## 5. Conclusions

In this study, we conclude that none of the four membrane impermeant CAIs had any effects on arteriolar wall tension. This suggests that CAIs may act as vasodilators through mechanisms located in the cytoplasm of vascular cells, although the precise mechanism of action remains to be elucidated.

## Data Availability

The data presented in this study are available upon request from the corresponding author.

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
