# Peer review of "Membrane Permeability Is Required for the Vasodilatory Effect of Carbonic Anhydrase Inhibitors in Porcine Retinal Arteries"

_ijms, 2023, doi:10.3390/ijms24098140_

Round 1
Reviewer 1 Report
The manuscript provides a simple observation that several membrane impermeant CAIs had no significant effect on arteriolar wall tension, and then concluded that CAI act as vasodilators through cellular mechanisms located in the cytoplasm of vascular cells (in Abstract) and membrane permeability is required for the vasodilatory effect of carbonic anhydrase inhibitors (in Title). However, I think there is a huge gap between the observation and conclusions. Without a clear mechanism of these CAIs on the enzyme, it is difficult to tell the reason for the negative results. Furthermore, the poor binding affinity to cytoplasmic CAs (Table 1) questioned the function of these molecules as CAI. The study is too preliminary and speculative.
Author Response
These are overstatements from the reviewer and we are in total disagreement with his/her opinion. We think the study brings new insights regarding the possible isoforms involved in the vasodilatory effects of the CA inhibitors, a phenomenon not yet explained after many years since its discovery. Knowing that the mechanism involved is present in the cytoplasm of cells, and not outside cell membranes is an important step towards explanation. A CAI that has been found to enter cells (benzolamide) is the only compound that induces vasodilation in this study. The binding affinity of the compounds tested to cytoplasmic CAs is comparable to CAIs that we have previously shown are vasodilators, and that indeed enter cell membranes.
Reviewer 2 Report
The manuscript infers that membrane permeability is required for the vasodilatory effect of CAIs. They have taken 3 new CAIs to prove their hypothesis. However, the reviewer thinks that following comments should be addressed for the study to look complete.
Major:
1. What is the kinetics of inhibition of Benzolamide on the CA isoforms reported in the study? Along with AAZ, comparison with Benzolamide under the same experimental set up would be good. Also, please mention the membrane permeability of AAZ.
2. How is the compound 1 synthesized (nothing is mentioned about its synthesis procedure) and please write the structure of the compound to compare it with other three compounds mentioned in the study.
3. Since the four CAIs are claimed to be new, how did authors conclude that the CAIs are membrane impermeant? Also, what is the plausible explanation of the membrane impermeability of the four CAIs?
Minor:
1. In the Introduction, the rationale about synthesizing the CAIs should have been mentioned. Also, whether there is any relation between the structures of the CAIs and their membrane permeability/impermeability (for example presence of long hydrocarbon chain, or polar group etc), should have been discussed.
2. It is mentioned that hCA I and II isozymes are ubiquitously expressed, however, nothing is told about the origin of other two isozymes (cytosolic or membrane bound?).

Author Response
The manuscript infers that membrane permeability is required for the vasodilatory effect of CAIs. They have taken 3 new CAIs to prove their hypothesis. However, the reviewer thinks that following comments should be addressed for the study to look complete.
Major:
- What is the kinetics of inhibition of Benzolamide on the CA isoforms reported in the study? Along with AAZ, comparison with Benzolamide under the same experimental set up would be good. Also, please mention the membrane permeability of AAZ.
Reply: All of this has been published many times, and the reviewer should know the literature when he/she accepts to review a paper which is not in his/her research field. See for example: Supuran CT, Scozzafava A. Benzolamide is not a membrane-impermeant carbonic anhydrase inhibitor. J Enzyme Inhib Med Chem. 2004 Jun;19(3):269-73.
- How is the compound 1 synthesized (nothing is mentioned about its synthesis procedure) and please write the structure of the compound to compare it with other three compounds mentioned in the study.
Reply: The synthesis of compound 1, its CA inhibitory properties and X-ray crystal structures were reported again many years ago (see for example: Menchise V, De Simone G, Alterio V, Di Fiore A, Pedone C, Scozzafava A, Supuran CT. Carbonic anhydrase inhibitors: stacking with Phe131 determines active site binding region of inhibitors as exemplified by the X-ray crystal structure of a membrane-impermeant antitumor sulfonamide complexed with isozyme II. J Med Chem. 2005 Sep 8;48(18):5721-7; all human CA inhibition data with the compound are available in Supuran CT. Carbonic anhydrases: novel therapeutic applications for inhibitors and activators. Nat Rev Drug Discov. 2008 Feb;7(2):168-81.), Furthermore, this compound has been used in many physiological studies published in relevant journals, e.g., Science (see Rummer JL, McKenzie DJ, Innocenti A, Supuran CT, Brauner CJ. Root effect hemoglobin may have evolved to enhance general tissue oxygen delivery. Science. 2013 Jun 14;340(6138):1327-9.), not only this, but many others.
- Since the four CAIs are claimed to be new, how did authors conclude that the CAIs are membrane impermeant? Also, what is the plausible explanation of the membrane impermeability of the four CAIs?
Reply: The new compounds (they are new, it is not only that they are claimed to be new) are positively charged. Anybody knowing a bit of chemistry understands that. We showed many years ago in the papers mentioned above and also in this more recent one (Alterio V, Esposito D, Monti SM, Supuran CT, De Simone G. Crystal structure of the human carbonic anhydrase II adduct with 1-(4-sulfamoylphenyl-ethyl)-2,4,6-triphenylpyridinium perchlorate, a membrane-impermeant, isoform selective inhibitor. J Enzyme Inhib Med Chem. 2018 Dec;33(1):151-157.) that positively charged compounds are membrane-impermeant.
Minor:
- In the Introduction, the rationale about synthesizing the CAIs should have been mentioned. Also, whether there is any relation between the structures of the CAIs and their membrane permeability/impermeability (for example presence of long hydrocarbon chain, or polar group etc), should have been discussed.
Reply: this has been added. There are no long alkyl chains in these molecules, as rather obvious from Scheme 1, whereas the presence of the bis-guanide moiety induces highly polar features, and the possibility to be doubly protonated at the physiological pH (at the two guanidine moieties)
- It is mentioned that hCA I and II isozymes are ubiquitously expressed, however, nothing is told about the origin of other two isozymes (cytosolic or membrane bound?).
Reply: these issues are rather well-known for people working in the CA field. Anyhow, we added a phrase in the text.
Reviewer 3 Report
Review comments on ijms-2254178
Journal:International Journal of Molecular Sciences
Manuscript ID: ijms-2254178
Type of manuscript: Communication
Title:Membrane permeability is required for the vasodilatory effect of carbonic anhydrase inhibitors in porcine retinal arteries
Authors:Thor Eysteinsson*, Andrea García-Llorca, Arnar Oessur Hardarson, Daniela Vullo, Fabrizio Carta, Claudiu T. Supuran
Submitted to section: Biochemistry
Major comments
It has been well established that a broad range of carbonic anhydrase inhibitors (CAIs)can act as vasodilators on pre-contracted retinal arterioles. These include CAIs that are used clinically as pressure-lowering drugs to treat glaucoma as well as newer compounds with different molecular structures and different affinity for the different CA isoforms. Although neither selective affinity for specific isoforms nor lipophilicity can fully determine the potency of CAIs as vasodilators, the evidence suggests that cytosolic isoforms of carbonic anhydrase or cytosolic binding sites of membrane bound isoforms play a critical role in mediating the effects of carbonic anhydrase on vascular wall tension.
Previously, Supuran and his coworkers found that benzolamide (BZA) is not a membrane-impermeant CAI but can enter red blood cells (Supuran & Scozzafava, 2004, J. Enz. Inhib. Med. Chem., 19, 269-273). Further, more recently, they found that CAIs with different structures (Eysteinsson et al, 2019, Int. J. Mol. Sci., 20, 467) and lipophilicity (Eysteinsson et al, 2022, Curr. Eye. Res., 47, 1615-1621) can act as potent vasodilators on isolated, pre-contracted segments of retinal arteries. On the other hand, a group of carbonic anhydrase inhibitors (CAIs) that cannot enter cell membranes have no significant effect on the wall tension of pre-contracted pig retinal arterial segments.
This study is an extension of their previous study (Eysteinsson et al, 2019, Int. J. Mol. Sci., 20, 467) proposing that the membrane permeability is required for the vasodilatory effect of CAIs using porcine retinal arteries. As a short communication paper, present result is very interesting and the manuscript is well written. The present result is consistent with their previous study and will attract many readers working in this field. Accordingly, the manuscript is acceptable for publication in Int. J. Mol. Sci. in the present form.

Author Response
Major comments
It has been well established that a broad range of carbonic anhydrase inhibitors (CAIs) can act as vasodilators on pre-contracted retinal arterioles. These include CAIs that are used clinically as pressure-lowering drugs to treat glaucoma as well as newer compounds with different molecular structures and different affinity for the different CA isoforms. Although neither selective affinity for specific isoforms nor lipophilicity can fully determine the potency of CAIs as vasodilators, the evidence suggests that cytosolic isoforms of carbonic anhydrase or cytosolic binding sites of membrane bound isoforms play a critical role in mediating the effects of carbonic anhydrase on vascular wall tension.
Previously, Supuran and his coworkers found that benzolamide (BZA) is not a membrane-impermeant CAI but can enter red blood cells (Supuran & Scozzafava, 2004, J. Enz. Inhib. Med. Chem., 19, 269-273). Further, more recently, they found that CAIs with different structures (Eysteinsson et al, 2019, Int. J. Mol. Sci., 20, 467) and lipophilicity (Eysteinsson et al, 2022, Curr. Eye. Res., 47, 1615-1621) can act as potent vasodilators on isolated, pre-contracted segments of retinal arteries. On the other hand, a group of carbonic anhydrase inhibitors (CAIs) that cannot enter cell membranes have no significant effect on the wall tension of pre-contracted pig retinal arterial segments.
This study is an extension of their previous study (Eysteinsson et al, 2019, Int. J. Mol. Sci., 20, 467) proposing that the membrane permeability is required for the vasodilatory effect of CAIs using porcine retinal arteries. As a short communication paper, present result is very interesting and the manuscript is well written. The present result is consistent with their previous study and will attract many readers working in this field. Accordingly, the manuscript is acceptable for publication in Int. J. Mol. Sci. in the present form.
Reply: We thank the reviewer for his/her positive statements on our work.
Round 2
Reviewer 1 Report
The authors did not solve my concerns about this manuscript. So I insist to think this manuscript is very preliminary and should not be published.
Furthermore, I also checked the authors’ responses to other reviewers’ comments. I think it is the authors’ responsibility to make the paper understandable to reviewers and readers (who are from much broader research fields than reviewers). The authors should not expect the reviewer and readers to have thorough background knowledge about CAIs. IJMS has broad readers from many subjects and this journal is not a specific journal in the CAI field.
Reviewer 2 Report
None.